# Therapeutic potential of vitamin D against bisphenol A-induced spleen injury in Swiss albino mice

**Mohamed A. Al-Griw[1], Hanan N. Balog[2], Taher Shaibi🄌[2], Mohamed Fouzi Elmoaket[3], Iman Said Ali AbuGamja[3], Ahlam Bashir AlBadawi[3], Ghalia Shamlan[4], Ammar Alfarga[5], Areej A. Eskandrani[5], Afnan M. Alnajeebi[6], Nouf A. Babteen[6], Wafa S. Alansari[6], Rabia Alghazeer[7]***

1 Department of Histology and Genetics, Faculty of Medicine, University of Tripoli, Tripoli, Libya, 2 Department of Zoology, Faculty of Sciences, University of Tripoli, Tripoli, Libya, 3 Tripoli Medical Center, Hematology Department, University of Tripoli, Tripoli, Libya, 4 Department of Food Science and Nutrition, College of Food and agriculture Sciences, King Saud University, Riyadh, Saudi Arabia, 5 Chemistry Department, Faculty of Science, Taibah University, Medina, Saudi Arabia, 6 Biochemistry Department, Faculty of Science, University of Jeddah, Jeddah, Saudi Arabia, 7 Department of Chemistry, Faculty of Sciences, University of Tripoli, Tripoli, Libya

* R.Alghazeer@uot.edu.ly

**Data Availability Statement:** All relevant data are within the paper and its Supporting information files.

## Abstract

Bisphenol A (BPA), a ubiquitous plasticizer, is capable of producing oxidative splenic injury, and ultimately led to spleen pathology. Further, a link between VitD levels and oxidative stress was reported. Hence the role of VitD in BPA-induced oxidative splenic injury was investigated in this study. Sixty male and female Swiss albino mice (3.5 weeks old) were randomly divided into control and treated groups 12 mice in each (six males and six females). The control groups were further divided into sham (no treatment) and vehicle (sterile corn oil), whereas the treatment group was divided into VitD (2,195 IU/kg), BPA (50 μg/kg), and BPA+VitD (50 μg/kg + 2,195 IU/kg) groups. For six weeks, the animals were dosed intraperitoneally (i.p). One week later, at 10.5 weeks old, mice were sacrificed for biochemical and histological analyses. Findings showed BPA triggered neurobehavioral abnormalities and spleen injury with increased apoptotic indices (e.g. DNA fragmentation) in both sexes. A significant increase was found in lipid peroxidation marker, MDA in splenic tissue, and leukocytosis. Conversely, VitD treatment altered this scenario into motor performance preservation, reducing oxidative splenic injury with a decrease in the percent apoptotic index. This protection was significantly correlated with preserving leukocyte counts and reduced MDA levels in both genders. It can be concluded from the above findings that VitD treatment has an ameliorative effect on oxidative splenic injury induced by BPA, highlighting the continuous crosstalk between oxidative stress and the VitD signaling pathway.

## Introduction

There are increasing evidence that environmental exposures during intrauterine and postnatal periods or early life determine the phenotypic outcome and vulnerability to diseases in later

**Funding:** The author(s) received no specific funding for this work.

**Competing interests:** The authors have declared that no competing interests exist.

life [1–3]. An endocrine disruptor, bisphenol A (BPA) has been used to produce epoxy resins and polycarbonate plastics [4]. BPA is essentially used for coating food cans and plastic food or beverage containers, resulting in regular exposure to humans [5, 6]. Therefore, continuous BPA exposure [5] is a severe cause of concern due to its detrimental effects on different body organs [6]. BPA exposure interferes with human health, leading to immune perturbations [4–7].

Spleen is an essential component of the immune system and performs vital functions, such as removal of degenerated and aged RBCs, extramedullary hematopoiesis, and elimination of particulate matter and harmful bacteria from the circulation [8, 9]. Because of its crucial role in the immune system, the spleen serves as an ideal organ for evaluating the effect of environmental toxicant BPA that interfere with normal hormone regulation of the immune response [6, 8, 10]. Therefore, the etiologies that dysregulate normal functions of the spleen also impair immune performance [6, 7, 8, 11, 12].

A lack of the underlying molecular mechanism that explains the diverse and pleiotropic impact on human health following BPA exposure led to further studies. Evidence suggests that lifestyle factors and exposure to environmental toxicants during developmental stages contribute to the generation of reactive oxygen species (ROS) and cause oxidative injury [10]. ROS-induced protein modifications are the underlying cause of several diseases [13]. The ROS, including the superoxide anion and hydroxyl radicals, impair DNA repair, enzyme activities, and oxygen utilization and depletes glutathione levels. Protein modifying effects of ROS leads to the production of lipid peroxidation-derived aldehydes and carbonyls such as 4-hydroxynonenal (HNE)-protein adducts and malondialdehyde (MDA). ROS-modified proteins can elicit an autoimmune response, which may result in the development of autoimmune disorders [8]. Furthermore, elevated protein carbonyls and MDA levels were found in patients with autoimmune disease [10, 12, 14]. Overproduction of ROS due to xenobiotic exposure leads to oxidative stress and its harmful consequences [15]. Oxidative stress is an underlying mechanism that can explain the correlation between BPA exposure and its adverse health effects [2, 5].

Vitamin D (VitD) is an integral part of immune system regulation and electrolyte reabsorption [16]. VitD2 and D3 are the two primary variants of VitD [16]. The 1,2,5-dihydroxy VitD3, also known as cholecalciferol, is the hormonally active form of VitD, vital in bone metabolism and calcium homeostasis [16]. Recently, a negative correlation between VitD levels and BPA exposure was reported [17–19]. A link between VitD levels and biomarkers related to oxidative stress and inflammation was suggested [20–22]. Although it plays a beneficial role in many biological processes [22], the protective role of VitD against oxidative response mediated spleen injury remains to be elucidated.

To the best of our knowledge, there is no/little data on the effects of VitD on BPA-induced oxidative splenic injury, therefore, therefore, we aim to investigate the possible protective role of VitD in BPA-induced oxidative splenic injury.

## Materials and methods

### Animals

Sixty male and female Swiss albino mice (3.5 weeks old) weighing approximately 17.1±1.8g were obtained from the Faculty of Sciences, University of Tripoli, Tripoli, Libya. All ethical regulations set by the Research Ethics Committee at Biotechnology Research Center, University of Tripoli, Libya, were followed to conduct animal work. Next, ethical approval was obtained from the Bioethics Committee at the Biotechnology Research Center before the commencement of the study (BEC-BTRC 10–2020). Every possible effort and precaution were taken to minimize pain to the animals throughout the experimental procedures. Animals were

maintained in a well-ventilated facility following a 12-hour light/dark cycle and ambient temperature (26 ± 2˚C). All animals received a standard diet and drinking water *ad libitum*. The diet was purchased from Research Diet Inc.

## Experimental design

The animals were randomly divided into control and treated groups 12 mice in each (six males and six females). The control groups were further divided into sham (no treatment) and vehicle (sterile corn oil), whereas the treatment group was divided into VitD (2,195 IU/kg), BPA (50 μg/kg), and BPA+VitD (50 μg/kg + 2,195 IU/kg) groups. The BPA and VitD doses were administered based on previously recommended doses [23–27]. Corn oil, VitD, and BPA were administrated daily, intraperitoneally (i.p), for six weeks. One week later, at 10.5 weeks old, animals were sacrificed, and blood and spleen samples were collected for biochemical and histological examinations (Fig 1).

The mice were administered intraperitoneally with 50 μg VitD, 2195 IU BPA/kg body weight, either alone or in combination, respectively, for six weeks.

## Clinical assessment

Survival of animals was measured throughout the study, mid-morning and late afternoon, for behavioral and clinical changes. Additionally, the death of mice occurring overnight was documented the following day, and three independent observers assessed the cause of the death to eliminate treatment-related deaths.

## Body and spleen weight

The body weight of mice at the beginning and end of the experiment was measured to determine changes in the body weight. The spleen was excised from the mice at the end of the treatment period, and to record any changes in spleen weight, their weights were compared to the spleen of control mice.

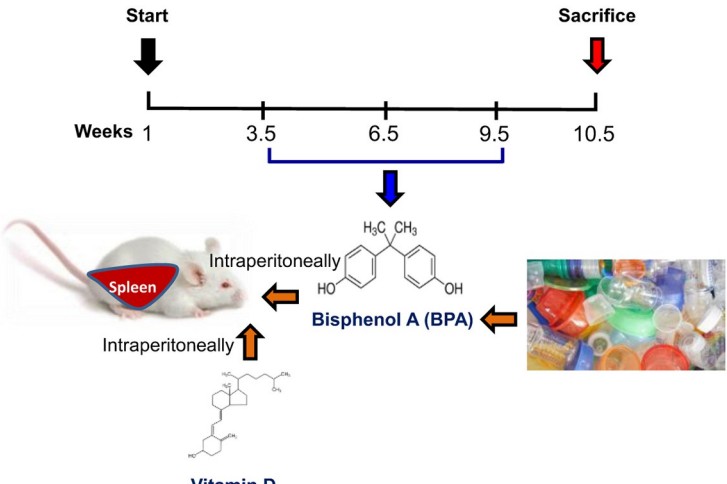

**Fig 1. Schematic of the treatment procedure.**

## Motor activity measurement

Motor performance was evaluated throughout the study by following a previously reported method [28, 29]. Briefly, the mice were subjected to regular training sessions on a rotating apparatus from week 4 to week 10. Motor performance was documented twice a week for six weeks. Three trials, each lasting for a minute, were carried out per session, and average values were calculated as the mean rotation number during five counting periods per hour.

## Peripheral blood and tissue harvesting

At 9.5 weeks old, the mice were anesthetized, and their blood samples were collected by cardiac puncture and placed in EDTA containing tubes. The animals were sacrificed under anesthesia with 1% ketamine. and their spleens were excised and weighed. A portion of the spleen was washed with the washing buffer and homogenized, and the supernatant obtained was used for lipid peroxidation assay. A portion of the splenic tissue was fixed in 10% formalin solution and stored for histological analysis.

## Leukocyte counting

The leukocytes (white blood cells) present in blood were viewed and counted on a blood film [7]. The blood film was air-dried, stained with Giemsa stain for 15 minutes, washed several times under tap water, air-dried, and mounted with Canada balsam. Differential cell counts were determined by observing the slides under an oil immersion microscope (Leica, Germany) using 40× magnification.

## Histopathology processing and analysis

The splenic tissue was fixed with 10% formalin for an hour after removal. Next, the slides were prepared according to the previously mentioned reports [7, 30]. The tissues were dehydrated by immersing overnight in 80% isopropyl alcohol for 60 minutes. The tissues were washed twice with xylene for an hour, and the wax-impregnated tissues were embedded in paraffin blocks, mounted, and 3-micron thick sections were cut using a microtome. The sections were floated in a tissue floatation bath at 40˚C and placed on egg albumin and glycerol-smeared glass slides, which were melted at 60˚C for 5 minutes and allowed to cool. Sections were deparaffinized in xylene for 10 minutes, washed in absolute isopropanol, and stained with Ehrlich's hematoxylin for 8 minutes. The excess stain was removed by washing the slides under tap water and immersing them in acid alcohol (8.3% HCl in 70% alcohol). The sections were washed for 10 minutes under running tap water to facilitate bluing (slow alkalization), were counter-stained for a minute in 1% aqueous eosin, and washed in tap water to remove excess stain. Next, the stained slides were dehydrated at 60˚C for 5 minutes, cooled, and mounted on a DPX mount. The sections were wetted in xylene and inverted onto a mount and placed on the coverslip for examination under a microscope.

 Tissue injury scores encompassing follicle degeneration, inflammation (macrophage), vascular congestion, and edema were established and compared among the study groups. On a scoring scale from 0 to 4, a score of 0 was assigned for no splenic tissue changes, 1 for little changes, 2 for moderate changes, 3 for marked changes, and 4 for very distinct changes. Above four, the scoring criteria were averaged for each section, and the average was considered a replicate. The tissue sections were examined under a light microscope (Leica, Germany) with lower and higher magnifications and imaged. A pathologist evaluated changes to tissue architecture.

### DNA integrity analysis

DNA was extracted using a commercial kit (Qiagen, Germany). Briefly, the tissue samples (25 mg) were minced and homogenized in a DNA lysis buffer containing proteinase K, followed by overnight incubation at 56˚C. The homogenate was treated with RNase A and loaded onto a spin column. The bound DNA was eluted using a Tris-EDTA (TE) buffer. The purity and integrity of the extracted DNA were determined by spectroscopy and agarose gel electrophoresis, respectively.

### Lipid peroxidation assay

The thiobarbituric acid reactive substances were used for measuring the malondialdehyde (MDA) levels [30–32]. Briefly, the splenic tissue was homogenized in phosphate-buffered saline using a tissue homogenizer (IKA, RW 20.n, Germany), and the homogenate obtained was centrifuged. A volume of 0.5 mL supernatant of the homogenate was mixed with a 2 mL reagent (0.37% thiobarbituric acid, 0.24 N HCl, and 15% TCA). The mixture was boiled at 100˚C for 15 minutes, cooled, and centrifuged at 3,000 rpm for 10 minutes. The absorbance of the supernatant was measured at 532 nm, and the unknown concentration of the TBA-MDA adducts was calculated from the standard curve generated using different concentrations of 1, 1, 3, 3-tetra methoxy propane and expressed as nmol/mL.

### Statistical analysis

Statistical analysis was carried out using the Graph Pad Prism software version 7.0. To determine statistically significant changes between male and female mice in all the groups, two-way ANOVA was followed by Dunnett's post hoc test. The normal distribution of the data was verified by skewness and kurtosis detection and homogeneity of variances by Levene's Test. Data are represented as mean ± SD (n = 12). Significance was set at $p < 0.05$.

## Results

### Survivability of mice

The clinical observations revealed no behavioral changes, mortality, or toxicity in all the groups in response to different treatments.

### Effect of VitD on motor activity

The results showed no difference between the treatment groups and the sham and vehicle control group (Fig 2).

However, an increase in motor activity was measured only in female mice treated with BPA compared to the females in the control group ($P = 0.03$; Fig 2). No such enhancement in motor activity was observed in the female mice treated with BPA+VitD. The activity was significantly decreased compared to the BPA-treated female mice group ($P = 0.0081$; Fig 2). No difference in motor activities was observed between other groups, and no sex-based difference in motor activities was noticed in the rest of the treatment groups.

### Effect of VitD and BPA on body and spleen weight

The changes in the body and spleen weight are presented in Fig 3A and 3B, respectively.

Compared to male and female control mice, the body weight of male mice treated with BPA was significantly ($P = 0.009$), while reduced in BPA-treated female mice ($P = 0.01$). Next, the body weight of BPA-treated female mice was significantly lower than BPA-treated male

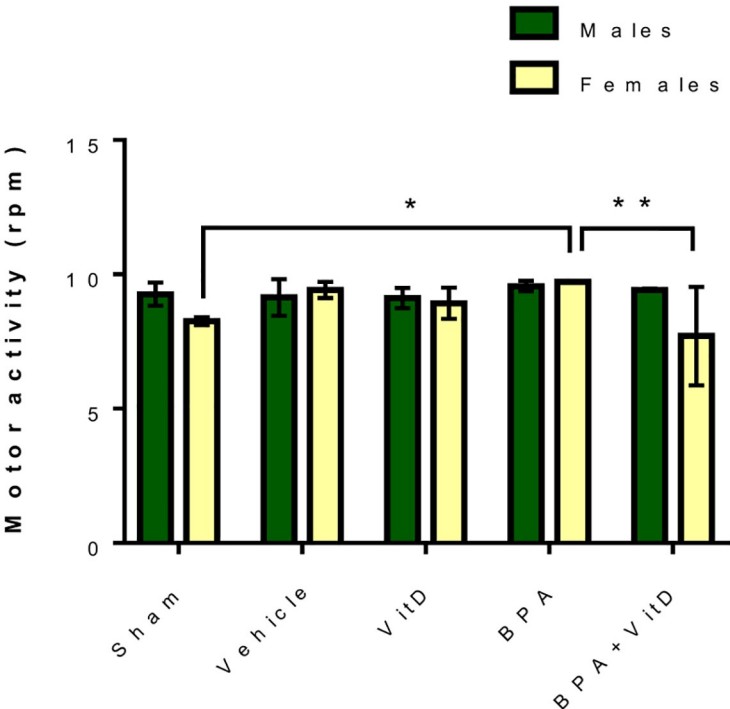

**Fig 2. Effect of BPA and VitD on the motor activity in male and female mice.** Data are represented as mean ± SD (n = 12), and P < 0.05 is represented as * and P < 0.01 is represented as **.

mice ($P = 0.0001$). Similarly, the body weight of female mice treated with BPA+VitD was significantly lower compared to the BPA+VitD treated male mice ($P = 0.0028$). The weight of the spleen was significantly increased in both male and female mice compared to their control counterparts ($P = 0.0001$). However, the weight of the spleen was unchanged in male mice treated with BPA+VitD compared to the weight of the spleen in BPA only treated male mice,

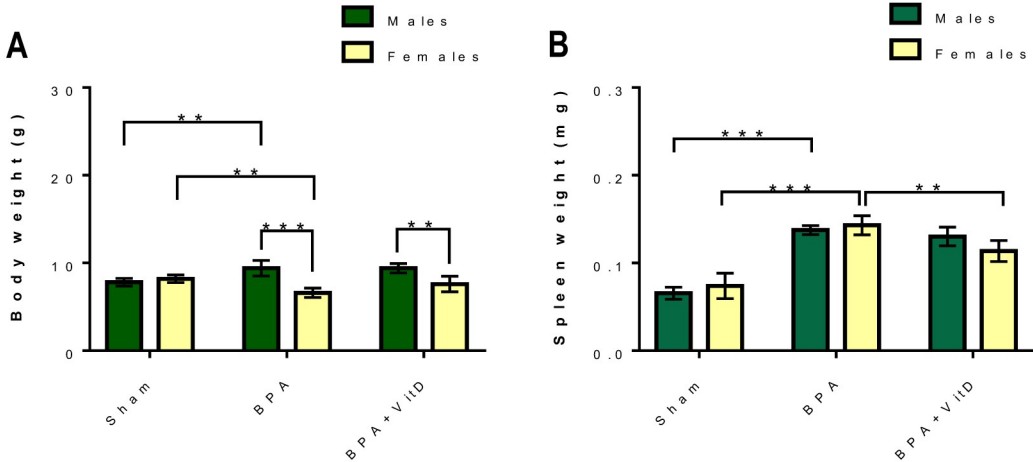

**Fig 3. Effect of VitD and BPA on mice body and spleen weight.** (A) Quantification of body weight, (B) Quantification of spleen weight. Data are represented as mean ± SD (n = 12), and P < 0.01 is represented as **, while P < 0.001 is represented as ***.

while it significantly decreased in female mice treated with BPA+VitD compared to the weight of the spleen in BPA only treated female mice ($P = 0.0021$). Next, no significant difference was found in spleen weight between BPA+VitD and BPA-treated male mice.

## VitD attenuates leukocyte count after BPA exposure

The data showed that the BPA-treated group decreased leukocyte (WBCs) counts compared to the control group (Fig 4A–4F).

In males, BPA increased the total WBCs ($P = 0.008$) and lymphocyte counts (Fig 4A and 4B) and reduced the neutrophil counts (Fig 4C). In females, BPA increased the total WBCs, monocytes, and neutrophils counts (Fig 4A–4C) and decreased the basophils and eosinophils counts (Fig 4D and 4F). Conversely, VitD treatment exerted favorable effects on the WBC counts (Fig 4). Specifically, VitD treatment enhanced the total WBC and monocyte counts in BPA male mice compared to the untreated BPA male mice (Fig 4A and 4B). While, VitD treatment modulated the total WBCs, monocytes, basophils, and eosinophils counts in BPA female mice compared to the untreated BPA-treated female mice (Fig 4A, 4B, 4D and 4F).

## VitD reduces spleen pathology induced by BPA exposure

The effect of VitD and BPA on spleen histology is represented in Fig 5.

Histological examination revealed a significant reversal of adverse effects of BPA in VitD-treated mice. VitD attenuated follicle degeneration, inflammation (macrophage), and vascular congestion (Fig 5). Splenic tissues of BPA+VitD-treated male and female mice demonstrated minimal degeneration and congestion to the splenic tissues of BPA-treated male and female mice. Similarly, a significant difference was found between male and female control groups (Figs 5–7).

Tissue injury scores are shown in Table 1. Follicle degeneration, inflammation (macrophage), and vascular congestion were compared among the different groups, and no significant difference was found in the mean tissue injury scores between the male and female control groups ($P < 0.05$; Table 1). The mean tissue injury scores were higher in the BPA-treated male and female mice compared to control ($P < 0.05$; Table 1). The BPA+VitD-treated male and female mice demonstrated significantly lower scores than BPA-treated mice. Based on these results, VitD significantly reduced tissue injury in the spleen of mice ($P < 0.05$; Table 1).

To understand the effect of treatment on cell necrosis in the splenocytes, we analyzed the total nuclear genomic DNA from the splenic tissues of mice by agarose gel electrophoresis (Fig 8A).

The results showed that DNA from the control group was largely intact and exhibited no/ little internucleosomal DNA fragmentation (Fig 8A, lane 1). However, the DNA from the male and female BPA-treated groups exhibited an apoptotic DNA ladder in gel electrophoresis (Fig 8A, lane 2). In contrast, the treatment with VitD enhanced the internucleosomal DNA integrity (Fig 8A, lane 3).

Quantitative analysis showed that treatment with BPA resulted in a decrease in the DNA concentration compared to the respective DNA concentration of the control group (Fig 8). Conversely, VitD treatment preserved the DNA integrity compared to the BPA treatment group (Fig 8B). No significant difference in the DNA integrity and quantity was observed between the adult males and females in all experimental groups.

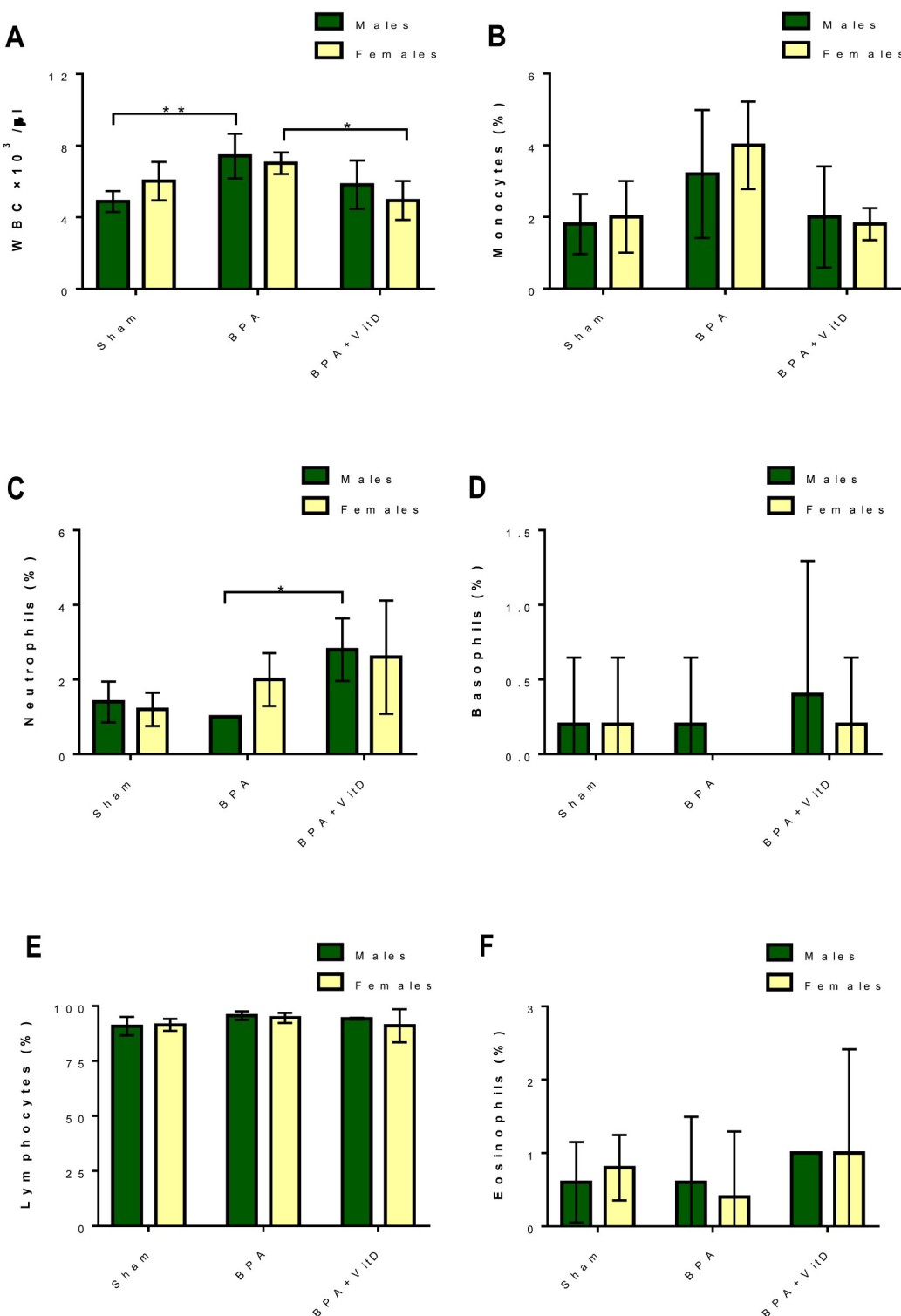

**Fig 4. VitD attenuates leukocyte counts following BPA exposure.** The mice were divided into different groups: control, vehicle, VitD, BPA, or BPA+VitD. The quantification of (A) WBCs, (B) monocytes, (C) neutrophils, (D), basophils, (E) lymphocytes, and (F) eosinophils. Data are represented as mean ± SD (n = 12), P < 0.05 is represented as *, and P < 0.01 is represented as **.

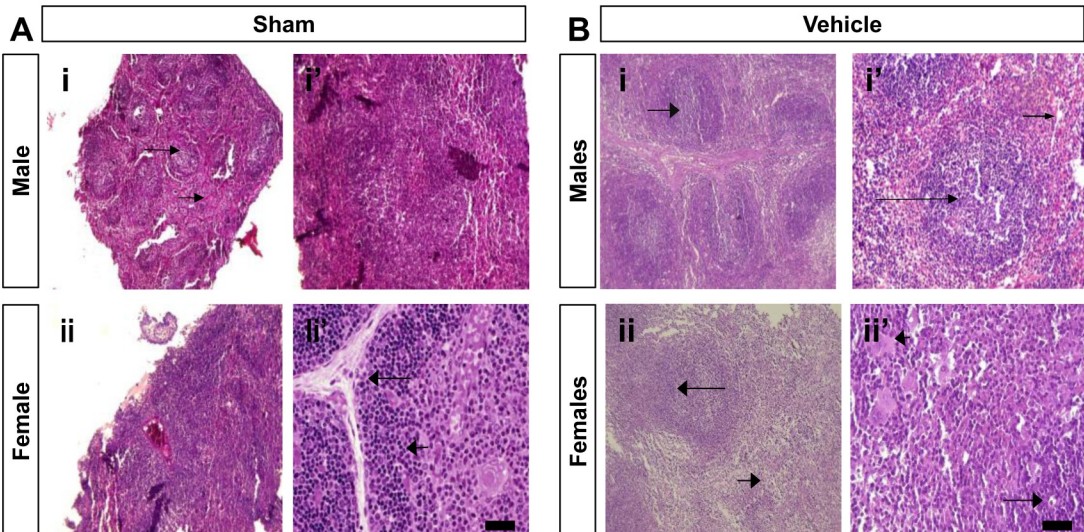

**Fig 5. Histopathological analysis in hematoxylin and eosin (H&E) stained spleen of control and vehicle-treated mice (40×).**
(A) The splenic tissues of male and female control mice at lower magnification (panels Ai and ii) showed a typical structure of the spleen. The lymphatic nodules showed a prominent germinal center composed of the white pulp (long arrow) surrounded by splenic cords and venous sinuses composed of the red pulp (short arrow). The splenic tissues of male and female control mice (panels Ai' and ii') showed splenic trabeculae (long arrow) and lymphocytic aggregation and part of white pulp in the lymphatic nodules (short arrow). (B) The splenic tissue of male and female vehicle-treated mice at lower magnification (panels Bi and ii) showed lymphatic nodules with a prominent germinal center composed of the white pulp (long arrow). Also, at higher magnification, the splenic tissues of male and female vehicle-treated mice (panels Bi' and ii') showed white pulp composed of lymphatic nodules and prominent germinal center (long arrow) and red pulp composed of splenic cords and venous sinuses (short arrow).

## VitD reduces lipid peroxidation in the spleen after BPA exposure

Herein, we analyzed the effect of BPA on oxidative stress in the splenic tissues by assessing levels of MDA, a marker of lipid peroxidation. Also, we studied the possible potential role of VitD in protecting the splenic tissue architecture against oxidative lipid damage. Initially, we found that BPA augmented the splenic MDA levels in males and females compared to the MDA levels of the control group ($P < 0.0001$ and $P < 0.0001$, respectively; Fig 9).

Conversely, VitD significantly decreased the MDA levels in the splenic tissues of females but not males compared to the MDA levels in the untreated female BPA group ($P = 0.0021$; Fig 9). No significant difference was found in the oxidative stress biomarker, MDA, observed between males and females in all experimental groups.

## Discussion

The current work demonstrated that treatment with BPA markedly altered the motor activity and body weight, decreased the spleen weight, increased splenic tissue injury scores, and ultimately led to spleen pathology. Also, a marked increase was found in lipid peroxidation. VitD treatment significantly reversed the pathological changes. To the best of our knowledge, this is the first study to demonstrate that VitD abrogates oxidative stress-induced splenic injury after treatment with BPA in mice. Our findings suggest that a VitD can act as a therapeutically potent agent in reversing BPA-induced pathophysiological changes.

Environmental exposure to BPA is a significant threat to public health worldwide. Several factors like environmental toxicants, unhealthy diet, or stress can promote different pathologies [30, 33–36]. Studies indicated a strong link between exposure to environmental toxicants,

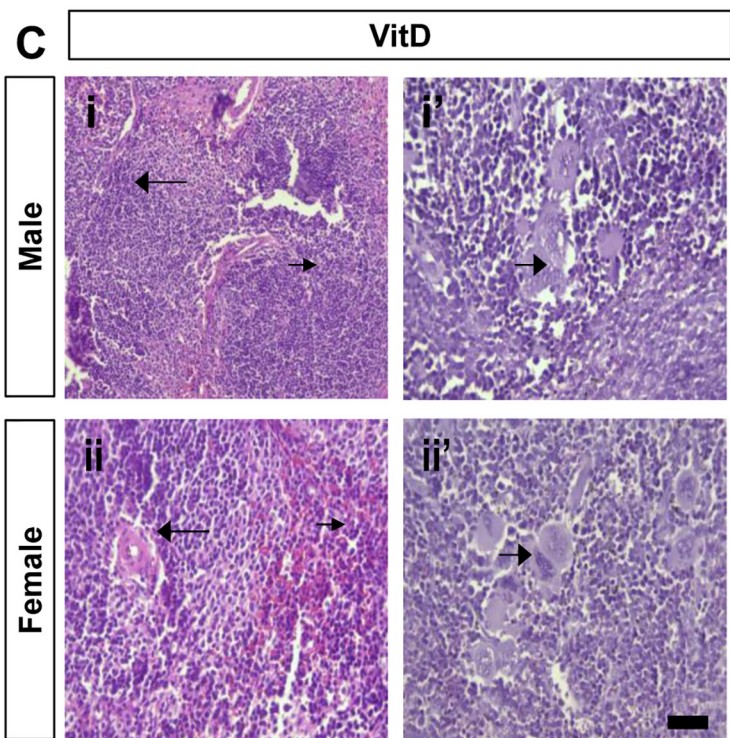

**Fig 6. Histopathological analysis in hematoxylin and eosin (H&E) stained spleen of the VitD-treated mice (40 X).**
The splenic tissues of male and female VitD-treated mice at high (panels Ci and ii) and low (panels Ci' and ii')
magnification exhibit a part of the reactive germinal center (long arrow) with congested sinusoids (long arrow) and a
few scattered macrophages (short arrow).

immune perturbations, [4, 5, 37]. BPA is a potent environmental toxicant that can negatively
affect the immune system even at lower doses [4, 38–41].

VitD plays a vital role in maintaining $Ca^{2+}$ homeostasis and bone metabolism [16, 22]. The
beneficial effects of VitD significantly augment the protective function of the innate immune
system [23]. VitD and its receptor activator have recently been shown to exert a protective
effect in ischemia and reperfusion injury [42]. Also, $VitD_3$ reversed kidney damage caused by
reperfusion injury [43]. Similarly, paricalcitol, the VitD receptor activator, restored renal
ischemia-reperfusion-induced damages [44]. Given the importance of VitD in normal physiol-
ogy, we sought to assess the possible protective efficacy of VitD against BPA-induced spleen
pathology.

In this study, BPA was found to trigger motor abnormalities in males and females, and that
co-treatment of BPA and VitD led to a significant reduce in BPA-induced motor abnormalities
[45–48]. These findings are consistent with several previous reports, wherein BPA was shown
to cause developmental motor abnormalities, and VitD treatment negatively correlated with
the risk of motor abnormalities [49].

Exposure to BPA during developmental stages, particularly in the pre or postnatal stages,
was correlated with increased body weight [6, 8, 50–53]. Notably, the effect of BPA on body
weight was reported to be gender-independent [6, 8]. In contrast, BPA at a dose concentration
of 0.1 mg/kg and $\geq$466 mg/kg/day significantly decreased the body weight compared to the
body weight of untreated control mice [54]. Similarly, an absolute (>22%) and relative
(>10%) decrease in the weight of rat livers were reported after exposure to BPA [55, 56]. Our

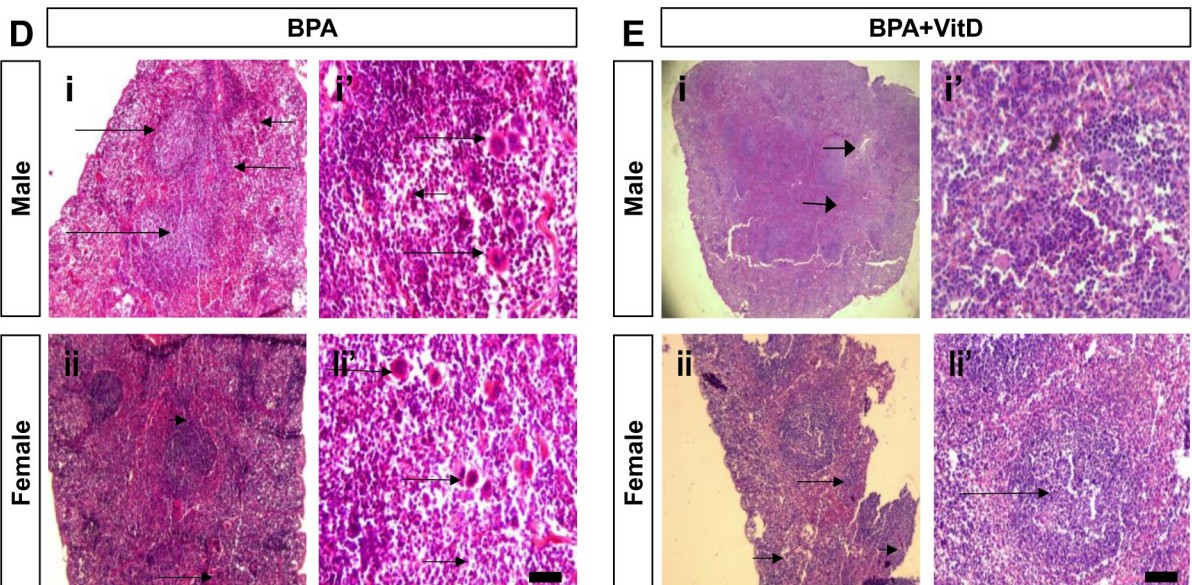

**Fig 7. Histopathological analysis in hematoxylin and eosin (H&E) stained spleen of the BPA-treated mice and BPA+VitD-treated mice (40×).** (D) The splenic tissues of male and female BPA-treated mice at lower magnification (panels Di and ii) showed hyperplasia of the splenic lymphoid follicles within the white pulp (long arrow), and the red pulp showed dilatation and congestion of the splenic blood vessels (short arrow). At lower magnification, the splenic tissues of male and female BPA-treated mice (panels Di' and ii') showed significant hyperplasia of the splenic lymphoid follicles (long arrow) surrounded by marked red pulp congestion (short arrow) and macrophage accumulation at the periphery of the tissues (long arrow) and massive hemosiderosis (short arrow). (E) The splenic tissues of male and female BPA+VitD-treated mice at lower magnification (panels Ei and ii) revealed hyperplasia of the splenic lymphoid follicles, surrounded by congested sinusoids with macrophage accumulation. At higher magnification, the splenic tissues of male and female BPA+VitD-treated mice (panels Ei' and ii') showed marked sinusoidal congestion with macrophage accumulation.

findings confirm that exposure to BPA changed the body and spleen weight in adult male and female mice. The role of VitD on body weight modulation has been controversial. BPA exposure for six weeks decreased the body and spleen weights but to non-significant levels, which supports a previous study, where no change in the body weight was noted in response to VitD [57].

Blood components control the immune response kinetics [58]. Leukocytes or WBCs are the body's primary defense against pathogenic challenges, which lead to antibody production and leukocytosis. Polymorphonuclear leukocytes with granular cytoplasm impart protection by releasing histamines. They lead to phagocytosis against invading pathogens and antigens, leading to inflammation [58]. The immune response defense is curtailed by exposure to BPA, leading to the depletion of eosinophils, basophils, and neutrophils [59] and impairment of WBC

**Table 1. Comparison of injury scores between different treatment groups.**

| Criteria of scoring | Control | | Vehicle | | VitD | | BPA | | BPA+VitD | |
|---|---|---|---|---|---|---|---|---|---|---|
| | Male | Female | Male | Female | Male | Female | Male | Female | Male | Female |
| Follicle degeneration | 1 | 1 | 1 | 1 | 1 | 1 | 3 | 3 | 2 | 2 |
| Inflammation (macrophage) | 1 | 1 | 1 | 1 | 1 | 1 | 4 | 4 | 3 | 3 |
| Vascular congestion | 0 | 0 | 0 | 0 | 1 | 1 | 4 | 4 | 3 | 3 |
| Averaged scores | 0.6 | 0.6 | 0.6 | 0.6 | 1 | 1 | 3.7 | 3.7 | 2.7 | 2.7 |

Data are represented as mean ± SD, n = 12 (six males and six females); Two-way ANOVA; VitD attenuates DNA damage in the spleens upon BPA exposure.

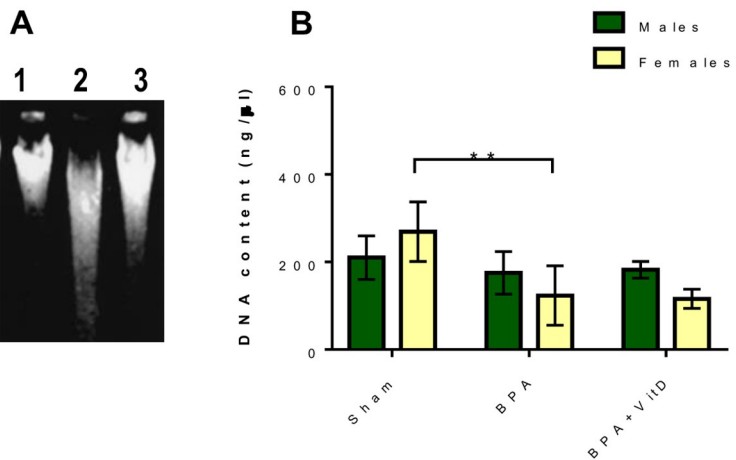

**Fig 8. VitD reduces the genomic DNA integrity in BPA-treated mice.** (A) Agarose gel electrophoresis of DNA isolated from the splenic tissues of control (lane 1), BPA-treated (lane 2), and BPA+VitD-treated (lane 3) mice. Data are represented as mean ± SD (n = 12), and P < 0.01 is represented.

production in the hematopoietic stem cells of the bone marrow [59]. The impairment of WBC production leads to BPA-induced oxidative stress [60]. In the present study, we found that BPA led to marked alterations in the WBC counts compared to the control. BPA induced a significant increase in the total WBC and lymphocyte counts in males, and reduced the monocyte, neutrophil, and basophil counts. In females, BPA induced a significant increase in the

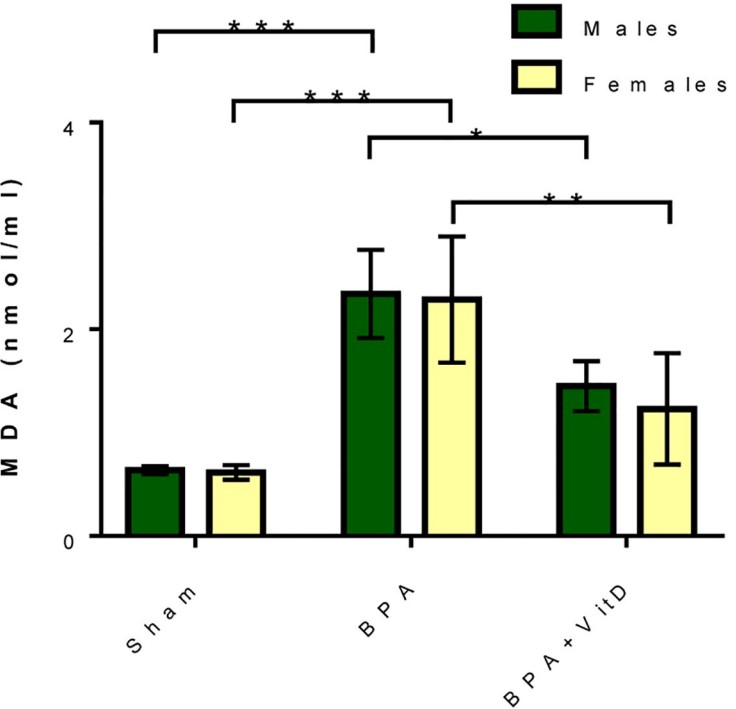

**Fig 9. VitD reduces BPA-induced lipid peroxidation.** Data are represented as mean ± SD (n = 12), and P < 0.05 is represented as *, P < 0.01 is represented as **, and P < 0.001 is represented as ***.

total WBC, monocyte, and lymphocyte counts and caused a reduction in the neutrophil, baso-phil, and eosinophil counts.

VitD was found to improve the hematological parameters in mice [57]. Moreover, research-ers reported that VitD enhances WBC counts [61]. Similarly, in this study, co-treatment with VitD reversed the effect of BPA on WBC counts. Specifically, VitD enhanced the counts of all WBC types in male mice compared to the counts in BPA-treated male mice. The co-treatment with VitD modulated the total WBC, neutrophil, and eosinophil counts but not monocyte and lymphocyte counts compared to the counts of BPA-treated female mice.

Increasing evidence documented that the dose and route of BPA administration determine its toxic effects. For example, BPA administration at 125 mg/kg/day via oral route demon-strated an adverse effect on the liver and kidneys [25, 62, 63]. Cellular and microanatomical abnormalities of the spleen and hematopoietic and immunomodulatory functions are affected by the dose of BPA and the sex of the animal [6]. BPA at a dose of 5 mg/kg/day exerted severe toxicity in the liver and kidneys, while BPA at 50 or 600 mg/kg/day failed to affect these organs [25, 62]. In the present study, we found that the treatment with BPA augmented the spleen pathology in male and female mice, including the cellular and microstructural alterations and depletion of lymphocytes in the white pulp accompanied by pyknotic cell aggregation. Also, narrow lumen, ruptured wall of central arterioles, changes to red pulp such as increased neu-trophil and macrophage counts, and nests of pyknotic cells were noted. Increased susceptibility of splenocytes to apoptosis may lead to spleen pyknosis, an underlying mechanism leading to immune senescence and autoimmune diseases [64]. Also, we found that DNA from the BPA-treated mice exhibited apoptotic DNA ladder in gel electrophoresis. The formation of the DNA ladder (180–200 bp fragments) can be analyzed by agarose gel electrophoresis [65]. Simi-lar to another study [66], increased apoptotic splenocytes were found in the BPA-treated male and female mice compared to control mice in the present investigation. VitD has been shown to prevent adverse effects of environmental toxins on spleen morphology [67] and ischemia-reperfusion injury of the ovary in animals [42]. Interestingly, improved spleen morphology and tissue architecture in VitD administrated mice were observed.

Several studies stressed the association between BPA exposure and the generation of oxida-tive stress and its role in the detrimental health effects [12]. Moreover, oxidative stress is the causal mechanism involved in the toxicity associated with BPA [5, 68–70]. Oxidative stress leads to several changes, including membrane injury, mitochondrial dysfunction, and DNA damage in splenocytes [66]. Thus, BPA-induced ROS generation and subsequent oxidative stress might be involved in toxicity [5]. In epidemiological studies, increased lipid peroxidation marker, MDA, was correlated with the BPA exposure, and in fact, its elevated levels were found in patients with autoimmune diseases [10]. To substantiate the effect of oxidative stress on BPA-induced spleen pathology, we measured the MDA levels in the splenic tissue homoge-nates in all experimental groups. Here, we found that BPA significantly increased the MDA levels in the splenic tissues of the adult male and female mice compared to their corresponding MDA levels in the control group.

Vitamins play a vital role in abrogating oxidative stress due to their antioxidant and anti-apoptotic properties. For instance, vitamin E supplementation attenuated apoptotic changes in the ovaries and caused reperfusion and ischemia [71]. VitD is a protective agent in many disor-ders [72, 73]. Many studies showed a strong correlation between VitD and oxidative stress [20–22, 74]. Researchers showed that ischemia and reperfusion-induced injury to the liver can be prevented with VitD supplementation [75]. Augmented MDA levels resulting from the non-alcoholic fatty liver disease were ameliorated after VitD supplementation [76]. In contrast, researchers found that VitD supplementation for twelve weeks reduced glutathione (GSH) and MDA levels while keeping the nitric oxide (NO) levels constant [77]. In this study, VitD

treatment led to a marked reduction in the BPA induced-oxidative stress marker, MDA, in the splenic tissues of mice, independent of their gender.

## Conclusion

Our findings led us to the conclusion that exposure to BPA caused splenic oxidative damage by affecting the status of oxidants and ultimately led to spleen pathology. Notably, the concurrent co-treatment with VitD demonstrated protective effects on the spleen architecture and WBC counts and afforded protection against BPA toxicity. However, this study fails to pinpoint the precise mechanism(s) VitD exerts its action. Nevertheless, the abovementioned data indicates that the favorable effects of VitD are mediated by its ability to block oxidative stress. Thus, VitD may serve as an effective treatment alternative for BPA-induced spleen injury. Further studies should be undertaken for a clear understanding of the protective mechanism of VitD.

## Supporting information

**S1 File.**
(PDF)

**S1 Raw images.**
(ZIP)

## Author Contributions

**Conceptualization:** Mohamed A. Al-Griw, Rabia Alghazeer.

**Data curation:** Hanan N. Balog, Taher Shaibi, Mohamed Fouzi Elmoaket, Iman Said Ali AbuGamja, Ahlam Bashir AlBadawi.

**Formal analysis:** Hanan N. Balog, Iman Said Ali AbuGamja, Ahlam Bashir AlBadawi, Ghalia Shamlan, Ammar Alfarga, Areej A. Eskandrani, Afnan M. Alnajeebi, Nouf A. Babteen, Wafa S. Alansari, Rabia Alghazeer.

**Investigation:** Mohamed A. Al-Griw, Taher Shaibi, Mohamed Fouzi Elmoaket, Rabia Alghazeer.

**Methodology:** Mohamed A. Al-Griw, Iman Said Ali AbuGamja, Ahlam Bashir AlBadawi, Rabia Alghazeer.

**Supervision:** Ghalia Shamlan, Ammar Alfarga, Areej A. Eskandrani, Wafa S. Alansari.

**Writing – original draft:** Mohamed A. Al-Griw, Hanan N. Balog, Taher Shaibi, Mohamed Fouzi Elmoaket.

**Writing – review & editing:** Mohamed A. Al-Griw, Ghalia Shamlan, Ammar Alfarga, Areej A. Eskandrani, Afnan M. Alnajeebi, Nouf A. Babteen, Wafa S. Alansari, Rabia Alghazeer.

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
