## [Decision Letter · Decision Letter 0]

31 Aug 2022

PONE-D-22-20374Therapeutic potential of vitamin D against bisphenol A-induced spleen injuryPLOS ONE

Dear Dr. Alghazeer,

Thank you for submitting your manuscript to PLOS ONE. After careful consideration, we feel that it has merit but does not fully meet PLOS ONE’s publication criteria as it currently stands. Therefore, we invite you to submit a revised version of the manuscript that addresses the points raised during the review process. Please submit your revised manuscript by Oct 15 2022 11:59PM. If you will need more time than this to complete your revisions, please reply to this message or contact the journal office at plosone@plos.org. Please include the following items when submitting your revised manuscript:A rebuttal letter that responds to each point raised by the academic editor and reviewer(s). You should upload this letter as a separate file labeled 'Response to Reviewers'.A marked-up copy of your manuscript that highlights changes made to the original version. You should upload this as a separate file labeled 'Revised Manuscript with Track Changes'.An unmarked version of your revised paper without tracked changes. You should upload this as a separate file labeled 'Manuscript'.

We look forward to receiving your revised manuscript.

Kind regards,

Yasmina Abd‐Elhakim

Academic Editor

PLOS ONE

Journal Requirements:

2. o comply with PLOS ONE submissions requirements, in your Methods section, please provide additional information regarding the experiments involving animals and ensure you have included details on (1) methods of sacrifice, (2) methods of anesthesia and/or analgesia, and (3) efforts to alleviate suffering.

5. Please ensure that you refer to Figures 1 and 7 in your text as, if accepted, production will need this reference to link the reader to the figure.

6. Please upload a copy of Figures A, B, D and F to which you refer in your text on page 9. If the figure is no longer to be included as part of the submission please remove all reference to it within the text.

Reviewers' comments:

Reviewer's Responses to Questions

**Comments to the Author**

1. Is the manuscript technically sound, and do the data support the conclusions?

Reviewer #1: Yes

Reviewer #2: Partly

Reviewer #3: Partly

2. Has the statistical analysis been performed appropriately and rigorously? 

Reviewer #1: Yes

Reviewer #2: Yes

Reviewer #3: Yes

3. Have the authors made all data underlying the findings in their manuscript fully available?

Reviewer #1: Yes

Reviewer #2: Yes

Reviewer #3: Yes

4. Is the manuscript presented in an intelligible fashion and written in standard English?

Reviewer #1: Yes

Reviewer #2: No

Reviewer #3: Yes

**5. Review Comments to the Author**

**Reviewer #1:** The authors addressed an important public health concern which is now a days quite common due to extensive use of disposable plastic in daily life. The role of Vit D to reverse the BPA induced changes is shown in the findings.

The study presented the findings of the original research and different sections like introduction, methods, results and discussion are well written and compiled. The data is analyzed using appropriate statistical tests.

The findings are concluded effectively. The authors reported that all required ethical approvals were sorted in advance from the relevant bodies.

The literature discussed in the introduction as well as in discussion section needs to be improved for the BPA and BPA related chemicals e.g.BPS etc. There have been some papers recently which can be taken into consideration. The DOI of the articles are https://doi.org/10.1155/2019/9292316 and https://doi.org/10.1007/s11356-022-21672-2

Please provide a justification or specific reason to study only the Spleen for the current study. Why they ignored liver as a matter of fact toxicant metabolism mostly occurs in liver. Please provide specific reason for eliminating or not considering the liver. Or if you can include liver in the study, it will for sure improve the quality of the manuscript.

Please clarify "DNA Genomic Analysis". Genomic analysis means totally different what has been presented in this article. The result shows the integrity of DNA but not the genomics. Therefore this section needs to be revised with specific language for the method used. Please also include what was the quantity of the DNA used for this test. How it was ensured the all the sample should carry equal amount of DNA?

The articles has been corrected for language by the professional services, however, the figures that are included in the manuscript needs revision and attention. The graphs are stretched without keep the constant ratio between width and height. The histopathological figures are stained good however, the photography needs improvement. The scale bars are not labeled to indicated what measurement the bar refers too, though, the captions are well elaborated. The figures are labeled with a group "SHAM" however no such group is elaborated in the text section. Please make the relevant corrections in the figures or the text so as the graphs and the text is in agreement when it comes to talk about different experimental groups.

In order to improve the quality of the manuscript and its pictorial reflection, the graphs and figures need revision.

**Reviewer #2:** Dear Editor, many thanks for your confidence for giving me the opportunity to review the manuscript entitled “Therapeutic potential of vitamin D against bisphenol A-induced spleen injury” with Manuscript number: PONE-D-22-20374 for the PLOS ONE Journal.

All comments are cited in the attached reviewed manuscript.

1- The rationale for the experimental design is not clear in both abstract and introduction. It should be very well clarified.

2- This manuscript needs language and grammar editing.

3- It is recommended to specify the species of study in the title

4- Line 34-35: Abstract can't stand alone, total number of animals used, group size, group names needed to be more illustrated.

5- Aim of the study is not clarified, authors doesn't illustrate that BPA induced spleen injury and trial to alleviate such detrimental effects. Authors specify that BPA alters Vit D thus, authors should illustrate why study spleen injury no other organs targeted by BPA. The main problem or study aim is not clarified.

6- Line 35: Why authors mentioned 1α, 25-(OH) 2D3 then mentioned Vit D, did they mean its Vit D metabolite or parent form. Write full term at first mention in the text then use abbreviations.

7- Line 39: authors didn't mention the age of mice at the start of the experiment , please illustrate what do you meant by this age

8- The BPA could enter organisms through the digestive tract, respiratory tract and skin absorption, of which the digestive tract is the largest source of absorption. Why BPA was administered intraperitoneally in current trial. Please illustrate why this route of exposure

9- Abstract, unfortunately can't stand alone, several points should be clarified for more illustration such as total number of experimental animals and groups. The measurements that done. Significance of the obtained results like those of lipid peroxidation, DNA fragmentation.

10- Conclusion in Abstract should be improved.

11- Introduction: Need more reviewing data about the toxic effects of the BPA regarding the spleen injury induced by BPA and not cover the study problem completely. Introduction should include some data refer to BPA and why authors particularly chose this model (mice in this age 3.5 weeks). I am confusing regarding the rationale and the aim of the study.

- Line 53-56, 69-71: these paragraphs is away from the aim, design of your current study. Did you investigate any developmental anomalies?. Authors should focus on the developmental period and age of mice used in the current trial.

- Authors are talking about oxidative stress and ROS without owing them to BPA exposure in previous studies

12- Material and methods section:

- Line 104: authors should mention total number of animals used, sex, and average body weights.

- Line 115-116: name of experimental groups should be clarified, the full name of the vehicle should be specified, Vit D mentioned into 2 different names in the abstract mentioned as 1α, 25-(OH) 2D3

- Authors mentioned in line 104 that mice were at age of 3.5 weeks then at line 122 at 10.5 weeks and the duration of exposure was 6 weeks which is different to the mentioned ages.

- It is very important to mention reference for oral LD50 that used in current study or your preliminary study citation.

- Line 123-124: For sample collection and preservation, please illustrate types of samples collected and the preservation method or type of preservatives used and for what measurements the samples are collected.

- For oxidative stress related indices why lipid peroxidation only measured what about other antioxidant enzymes, glutathione, protein carbonyl, total antioxidant capacity, and specific marker of oxidative DNA damage (8-hydroxy-2-deoxyguanosine)

- Line 145-147: authors mentioned that blood collected, left to clot to obtain serum. Does leukocytic count done in serum????????????. Or in whole blood also, if whole blood collected please mention type of anticoagulant used.

- For lipid peroxidation MDA measures in spleen homogenate? so, why serum is collected.

13- For results:

- Figure 1 is not cited in the text

- The mean values of the obtained results of all measurements should be mentioned in the result section.

- The obtained results, all the measured parameters should be presented as the difference between treatment groups then the difference between sexes.

14- Discussion section:

- Firstly, authors should discuss the observed changes in behavioral observations like increased motor activity

- Line 296-297 authors states that BPA decreased body and spleen weight which in conversely matched with those obtained results in line 227-229

- Not all the significantly obtained results are fully discussed

- After carefully reinvestigating their preliminary observations, the author could improve the quality of discussion better by integrating the results with the current literature for each discussed parameter.

- Conclusion is too long

**Reviewer #3: **The authors of the manuscript entitled “ Therapeutic potential of vitamin D against bisphenol A induced spleen injury” is well written and easy to understand. The manuscript could be considered for publication provided the authors can address major issues listed below.

1. In introduction the author mentioned about the negative effects of vitamin D, briefly explain the negative effects.

2. In material and methods author didn’t mention the detail information about the diet given to mice.

3. In statical analysis the author performed two way ANOVA analysis, a suggestion of performing one way ANOVA comparison within the groups

4. Why the author choose 50ug of BPA dose.

5. Mostly the BPA and Vitamin doses are administered orally, why the author choose to inject intraperitoneally.

6. The discussion is quiet long, it’s better to concise it.

6. PLOS authors have the option to publish the peer review history of their article (what does this mean?). If published, this will include your full peer review and any attached files.

---

## [Author Response · Author response to Decision Letter 0]

29 Sep 2022

PONE-D-22-20374

Therapeutic potential of vitamin D against bisphenol A-induced spleen injury in Swiss albino mice

Thanks for reviewers for their good efforts to examine our MS, and the following responses have been made regarding their comments: 

Journal Requirements:

Authoress’s Response

The manuscript has been adjusted to meet PLOS ONE's style requirements

2. comply with PLOS ONE submissions requirements, in your Methods section, please provide additional information regarding the experiments involving animals and ensure you have included details on (1) methods of sacrifice, (2) methods of anesthesia and/or analgesia, and (3) efforts to alleviate suffering.

Authoress’s Response

─ methods of sacrifice → The animals were sacrificed under anesthesia, and the 

─ methods of anesthesia and/or analgesia → with 1% ketamine for dissection

─ efforts to alleviate suffering → reduce no of animal used

Authoress’s Response

There are no legal or ethical restrictions

4. Review Comments to the Author

Reviewer #1: 

Thanks for reviewer 1 for the good effort to examine our MS.

1. The literature discussed in the introduction as well as in discussion section needs to be improved for the BPA and BPA related chemicals e.g.BPS etc. There have been some papers recently which can be taken into consideration. The DOI of the articles are https://doi.org/10.1155/2019/9292316 and https://doi.org/10.1007/s11356-022-21672-2

Authoress’s Response

The introduction was modified.

2. Please provide a justification or specific reason to study only the Spleen for the current study. Why they ignored liver as a matter-of-fact toxicant metabolism mostly occurs in liver. Please provide specific reason for eliminating or not considering the liver. Or if you can include liver in the study, it will for sure improve the quality of the manuscript.

Authoress’s Response

BPA exposure has detrimental effects on the vital organs in humans and animals. Since the spleen includes vascular and lymphoid components and is a major site for activating primary immune responses (Suttie, 2006; Dong et al., 2013), there is increased interest in ascertaining molecular mechanisms and mode of actions underlying spleen pathology induced by BPA. 

3. Please clarify "DNA Genomic Analysis". Genomic analysis means totally different what has been presented in this article. The result shows the integrity of DNA but not the genomics. Therefore, this section needs to be revised with specific language for the method used. Please also include what was the quantity of the DNA used for this test. How it was ensured the all the sample should carry equal amount of DNA?

Authoress’s Response

The DNA genomic analysis has been replaced with analysis of DNA integrity

4. The article has been corrected for language by the professional services, however, the figures that are included in the manuscript needs revision and attention. The graphs are stretched without keep the constant ratio between width and height. The histopathological figures are stained good however, the photography needs improvement. The scale bars are not labeled to indicated what measurement the bar refers too, though, the captions are well elaborated. The figures are labeled with a group "SHAM" however no such group is elaborated in the text section. Please make the relevant corrections in the figures or the text so as the graphs and the text is in agreement when it comes to talk about different experimental groups.

Authoress’s Response

The text has been adjusted.

5. In order to improve the quality of the manuscript and its pictorial reflection, the graphs and figures need revision.

Authoress’s Response

The graphs and figures have been revised.

Reviewer #2: 

Thanks for reviewer 2 for the good effort to examine our MS.

1. The rationale for the experimental design is not clear in both abstract and introduction. It should be very well clarified.

Authoress’s Response

The rationale for the experimental design has been revised.

2. This manuscript needs language and grammar editing.

Authoress’s Response

The manuscripts language has been edited.

3. It is recommended to specify the species of study in the title

Authoress’s Response

The title has been adjusted.

4. Line 34-35: Abstract can't stand alone, total number of animals used, group size, group names needed to be more illustrated.

Authoress’s Response

The title has been adjusted.

5. Aim of the study is not clarified, authors don’t illustrate that BPA induced spleen injury and trial to alleviate such detrimental effects. Authors specify that BPA alters Vit D thus, authors should illustrate why study spleen injury no other organs targeted by BPA. The main problem or study aim is not clarified.

Authoress’s Response

BPA exposure has detrimental effects on the vital organs in humans and animals. Since the spleen includes vascular and lymphoid components and is a major site for activating primary immune responses (Suttie, 2006; Dong et al., 2013), there is increased interest in ascertaining molecular mechanisms and mode of actions underlying spleen pathology induced by BPA. 

6. Line 35: Why authors mentioned 1α, 25-(OH) 2D3 then mentioned Vit D, did they mean its Vit D metabolite or parent form. Write full term at first mention in the text then use abbreviations.

Authoress’s Response

The word 1α, 25-(OH) 2D3 was replaced for consistency.

7. Line 39: authors didn't mention the age of mice at the start of the experiment, please illustrate what do you meant by this age

Authoress’s Response

The age of animals was included

8. The BPA could enter organisms through the digestive tract, respiratory tract and skin absorption, of which the digestive tract is the largest source of absorption. Why BPA was administered intraperitoneally in current trial. Please illustrate why this route of exposure

Authoress’s Response

In the current study we sought to explore efficacy of VitD against BPA-induced spleen injury in a mouse model as well as characterized the underlying molecular mechanism(s).

9. Abstract, unfortunately can't stand alone, several points should be clarified for more illustration such as total number of experimental animals and groups. The measurements that done. Significance of the obtained results like those of lipid peroxidation, DNA fragmentation.

Authoress’s Response

The abstract was revised.

10. Conclusion in Abstract should be improved.

Authoress’s Response

The abstract was revised.

Introduction: 

11. Need more reviewing data about the toxic effects of the BPA regarding the spleen injury induced by BPA and not cover the study problem completely. Introduction should include some data refer to BPA and why authors particularly chose this model (mice in this age 3.5 weeks). I am confusing regarding the rationale and the aim of the study.

Authoress’s Response

The rationale and the aim of the study was revised.

12. Line 53-56, 69-71: these paragraphs is away from the aim, design of your current study. Did you investigate any developmental anomalies? Authors should focus on the developmental period and age of mice used in the current trial.

Authoress’s Response

The text has been revised.

13. Authors are talking about oxidative stress and ROS without owing them to BPA exposure in previous studies

Authoress’s Response

The text has been revised.

Material and methods section:

14. Line 104: authors should mention total number of animals used, sex, and average body weights.

Authoress’s Response

Total number of animals used, sex, and average body weights were included.

15. Line 115-116: name of experimental groups should be clarified, the full name of the vehicle should be specified, Vit D mentioned into 2 different names in the abstract mentioned as 1α, 25-(OH) 2D3

Authoress’s Response

The word 1α, 25-(OH) 2D3 was replaced for consistency.

16. Authors mentioned in line 104 that mice were at age of 3.5 weeks then at line 122 at 10.5 weeks and the duration of exposure was 6 weeks which is different to the mentioned ages.

Authoress’s Response

The text was modified.

17. It is very important to mention reference for oral LD50 that used in current study or your preliminary study citation.

Authoress’s Response

The dose of BPA (50 μg/kg) was chosen based on previous studies (Sadowski et al., 2014) and Al-Griw et al., 2022 (doi: 10.5455/OVJ.2022.v12.i1.4).

18. Line 123-124: For sample collection and preservation, please illustrate types of samples collected and the preservation method or type of preservatives used and for what measurements the samples are collected.

Authoress’s Response

The sample types and preservation method were included (see Peripheral blood and tissue harvesting in the methods section).

19. For oxidative stress related indices why lipid peroxidation only measured what about other antioxidant enzymes, glutathione, protein carbonyl, total antioxidant capacity, and specific marker of oxidative DNA damage (8-hydroxy-2-deoxyguanosine)

Authoress’s Response

The Lipid peroxidation was only used as biomarkers for oxidative stress injury.

20. Line 145-147: authors mentioned that blood collected, left to clot to obtain serum. Does leukocytic count done in serum? Or in whole blood also, if whole blood collected please mention type of anticoagulant used.

Authoress’s Response

The EDTA containing tube was used to collect the blood.

21. For lipid peroxidation MDA measures in spleen homogenate? so, why serum is collected.

Authoress’s Response

The text was modified.

For results:

22. Figure 1 is not cited in the text

Authoress’s Response

The figure 1 was included in the text.

23. The mean values of the obtained results of all measurements should be mentioned in the result section.

Authoress’s Response

The all measurements have been included in the result section. 

The obtained results, all the measured parameters should be presented as the difference between treatment groups then the difference between sexes.

Authoress’s Response

The measured parameters have been included.

Discussion section:

24. Firstly, authors should discuss the observed changes in behavioral observations like increased motor activity

Authoress’s Response

The section related to observed alterations in motor activity has been revised.

25. Line 296-297 authors states that BPA decreased body and spleen weight which in conversely matched with those obtained results in line 227-229

Authoress’s Response

The text was modified.

26. Not all the significantly obtained results are fully discussed

Authoress’s Response

The significant findings have been discussed in the discussion section.

27. After carefully reinvestigating their preliminary observations, the author could improve the quality of discussion better by integrating the results with the current literature for each discussed parameter.

Authoress’s Response

The discussion section has been revised and improved.

28. Conclusion is too long

Authoress’s Response

The conclusion has been revised and reduced.

Reviewer #3:

Thanks for reviewer 3 for the good effort to examine our MS.

1. In introduction the author mentioned about the negative effects of vitamin D, briefly explain the negative effects.

Authoress’s Response

There is no mention to the VitD negative effect. We say “Recently, a negative correlation between VitD levels and BPA exposure was reported”

2. In material and methods author didn’t mention the detail information about the diet given to mice.

Authoress’s Response

All animals used in the current study received a standard diet and drinking water ad libitum.

3. In statical analysis the author performed two-way ANOVA analysis, a suggestion of performing one-way ANOVA comparison within the groups

Authoress’s Response

To determine statistically significant changes between the treatment groups and control within male and female rats, a two-way ANOVA with Dunnett’s posttest was used.

4. Why the author chooses 50ug of BPA dose.

Authoress’s Response

It is the daily safety dose. The dose of BPA (50 μg/kg) was chosen based on previous studies (Sadowski et al., 2014) and Al-Griw et al., 2022 (doi: 10.5455/OVJ.2022.v12.i1.4).

5. Mostly the BPA and Vitamin doses are administered orally, why the author choose to inject intraperitoneally.

Authoress’s Response

We administered the BPA and VitD intraperitoneally rather than orally to animals for shorter periods and the absorption rate is higher. 

6. The discussion is quiet long, it’s better to concise it.

Authoress’s Response

The discussion was revised and reduced.

---

## [Decision Letter · Decision Letter 1]

2 Nov 2022

PONE-D-22-20374R1

Therapeutic potential of vitamin D against bisphenol A-induced spleen injury

PLOS ONE

Dear Dr. Alghazeer,

Thank you for submitting your manuscript to PLOS ONE. After careful consideration, we feel that it has merit but does not fully meet PLOS ONE’s publication criteria as it currently stands. Therefore, we invite you to submit a revised version of the manuscript that addresses the points raised during the review process.

We look forward to receiving your revised manuscript.

Kind regards,

Yasmina Abd‐Elhakim

Academic Editor

PLOS ONE

Journal Requirements:

Additional Editor Comments:

The authors are highly recommended to carefully address all reviewers comment and give a response to each point.

Reviewers' comments:

Reviewer's Responses to Questions

**Comments to the Author**

1. If the authors have adequately addressed your comments raised in a previous round of review and you feel that this manuscript is now acceptable for publication, you may indicate that here to bypass the “Comments to the Author” section, enter your conflict of interest statement in the “Confidential to Editor” section, and submit your "Accept" recommendation.

Reviewer #1: All comments have been addressed

Reviewer #2: (No Response)

Reviewer #3: All comments have been addressed

2. Is the manuscript technically sound, and do the data support the conclusions?

Reviewer #1: Yes

Reviewer #2: Partly

Reviewer #3: Partly

3. Has the statistical analysis been performed appropriately and rigorously? 

Reviewer #1: Yes

Reviewer #2: Yes

Reviewer #3: Yes

4. Have the authors made all data underlying the findings in their manuscript fully available?

Reviewer #1: Yes

Reviewer #2: Yes

Reviewer #3: Yes

5. Is the manuscript presented in an intelligible fashion and written in standard English?

Reviewer #1: Yes

Reviewer #2: Yes

Reviewer #3: Yes

6. Review Comments to the Author

Reviewer #1: (No Response)

Reviewer #2: Authors have responded to majority of inquiries, but there are some reviewing comments authors didn't reply to them as following

1- The BPA could enter organisms through the digestive tract, respiratory tract and skin absorption, of which the digestive tract is the largest source of absorption. Why PSA was administered intraperitoneally in current trial. Please illustrate why this route of exposure

2- Line 69-81: the paragraph is talking about Ni not NiCl2, there is no any reference regarding the NiCl2 reproductive effects

3- Line 53-56, 69-71: these paragraphs is away from the aim, design of your current study. Did you investigate any developmental anomalies?. Authors should focus on the developmental period and age of mice used in the current trial.

4- Authors are talking about oxidative stress and ROS without owing them to BPA exposure in previous studies

Material and methods section:

- type of homogenate

- For oxidative stress related indices why lipid peroxidation only measured what about other antioxidant enzymes, glutathione, protein carbonyl, total antioxidant capacity, and specific marker of oxidative DNA damage (8-hydroxy-2-deoxyguanosine)

- For lipid peroxidation MDA measures in spleen homogenate? So, why serum is collected.

- citing reference of method for DNA integrity analysis

For results:

- It's very important to specify your obtained results such as body and spleen weight decreased or increased in PSA exposed male mice please revise again. Also mean values of the obtained results are not mentioned as required for all measurements

- The obtained results, all the measured parameters should be presented as the difference between treatment groups then the difference between sexes.

Discussion section:

- Firstly, authors should discuss the observed changes in behavioral observations like increased motor activity

- Line 296-297 authors states that BPA decreased body and spleen weight which in conversely matched with those obtained results in line 227-229

- After carefully reinvestigating their preliminary observations, the author could improve the quality of discussion better by integrating the results with the current literature for each discussed parameter.

Reviewer #3: The authors of the manuscript entitled “ Therapeutic potential of vitamin D against bisphenol A induced spleen injury” is well written and easy to understand. The manuscript could be considered for publication provided the authors can address major issues listed below.

• The author mentioned about the standard diet. Kindly include the components or company name from where they are getting the feed.

• The biological mechanism of males and females are quite different, why author choose to compare between males and females

• To support the date, it’s better to check the activity of phospho tyrosine and overall PKA specific kinase activity in spleen through western blot to identify certain mechanism involved.

7. PLOS authors have the option to publish the peer review history of their article (what does this mean?). If published, this will include your full peer review and any attached files.

Reviewer #1: No

Reviewer #2: **Yes: **Ehsan H. Abu-Zeid

Reviewer #3: **Yes: **Shehreen Amjad

---

## [Author Response · Author response to Decision Letter 1]

25 Nov 2022

20 November 2022

Manuscript ID: PONE-D-22-20374R1

Manuscript title: “Therapeutic potential of vitamin D against bisphenol A induced spleen injury”

Response to reviewers

Comment-1

• The author mentioned about the standard diet. Kindly include the components or company name from where they are getting the feed.

Response-1

The diet was purchased from Research Diet Inc.

Comment-2

• The biological mechanism of males and females are quite different, why author choose to compare between males and females. To support the date, it’s better to check the activity of phospho tyrosine and overall PKA specific kinase activity in spleen through western blot to identify certain mechanism involved.

Response-2

Of course, the biological mechanisms of mouse males and females are different and it was reported that they repose to the bisphenol A (BPA) differently, therefore, we sought in this work to reinvestigate the effect of BPA on both males and females as well as to explore the role of VitD in BPA-induced oxidative splenic injury.

---

## [Decision Letter · Decision Letter 2]

8 Jan 2023

Therapeutic potential of vitamin D against bisphenol A-induced spleen injury

PONE-D-22-20374R2

Dear Dr. Alghazeer,

We’re pleased to inform you that your manuscript has been judged scientifically suitable for publication and will be formally accepted for publication once it meets all outstanding technical requirements.

Kind regards,

Yasmina Abd‐Elhakim

Academic Editor

PLOS ONE

---

## [Editor Report · Acceptance letter]

27 Feb 2023

PONE-D-22-20374R2 

Therapeutic potential of vitamin D against bisphenol A-induced spleen injury in Swiss albino mice 

Dear Dr. Alghazeer:

I'm pleased to inform you that your manuscript has been deemed suitable for publication in PLOS ONE. Congratulations! Your manuscript is now with our production department. 

Kind regards, 

on behalf of

Dr. Yasmina Abd‐Elhakim 

Academic Editor

PLOS ONE